# Techno-Environmental Analysis of the Use of Green Hydrogen for Cogeneration from the Gasification of Wood and Fuel Cell

**Abigail Gonzalez-Diaz [1],\*, Juan Carlos Sánchez Ladrón de Guevara [2], Long Jiang [3],\*, Maria Ortencia Gonzalez-Diaz [4], Pablo Díaz-Herrera [2] and Carolina Font-Palma [5],\***

[1]  National Institute of Electricity and Clean Energy, Reforma 113, Col. Palmira, Cuernavaca Morelos 62490, Mexico
[2]  Facultad de Ingeniería, Universidad Nacional Autonoma de Mexico, Mexico City 04510, Mexico; juan77758@gmail.com (J.C.S.L.d.G.); pablor.diazh@gmail.com (P.D.-H.)
[3]  Institution of Refrigeration and Cryogenics, Zhejiang University, Hangzhou 310027, China
[4]  CONACYT—Centro de Investigación Científica de Yucatán, A.C., Calle 43 No. 130, Chuburná de Hidalgo, Mérida 97200, Mexico; maria.gonzalez@cicy.mx
[5]  Department of Engineering, University of Hull, Hull HU6 7RX, UK
\*  Correspondence: abigail.gonzalez@ineel.mx (A.G.-D.); jianglong@zju.edu.cn (L.J.); c.font-palma@hull.ac.uk (C.F.-P.)

**Abstract:** This paper aims to evaluate the use of wood biomass in a gasifier integrated with a fuel cell system as a low carbon technology. Experimental information of the wood is provided by the literature. The syngas is purified by using pressure swing adsorption (PSA) in order to obtain $H_2$ with 99.99% purity. Using 132 kg/h of wood, it is possible to generate 10.57 kg/h of $H_2$ that is used in a tubular solid oxide fuel cell (TSOFC). Then, the TSOFC generates 197.92 kW. The heat generated in the fuel cell produces 60 kg/h of steam that is needed in the gasifier. The net efficiency of the integrated system considering only the electric power generated in the TSOFC is 27.2%, which is lower than a gas turbine with the same capacity where the efficiency is around 33.1%. It is concluded that there is great potential for cogeneration with low carbon emission by using wood biomass in rural areas of developing countries e.g., with a carbon intensity of 98.35 kgCO$_2$/MWh when compared with those of natural gas combined cycle (NGCC) without and with CO$_2$ capture i.e., 331 kgCO$_2$/MWh and 40 kgCO$_2$/MWh, respectively. This is an alternative technology for places where biomass is abundant and where it is difficult to get electricity from the grid due to limits in geographical location.

**Keywords:** biomass; hydrogen; fuel cell; gasification; pressure swing adsorption; carbon intensity

## 1. Introduction

The 2 °C goal set by the Paris climate agreement places rigorous limits on greenhouse gas (GHG) emission. Most climate and integrated assessment models project that the concentration of CO$_2$ in the atmosphere would have to decrease by the second half of the 21st century to achieve the 2 °C target [1]. In this sense, the deployment of negative emission technologies (NETs) becomes a key mitigation tool, which combines the production of energy from plant biomass to produce electricity. There are different NET technologies, the most important being [1]: e.g., coastal blue carbon (CBC), bioenergy with carbon capture and sequestration (BECCS), and direct air capture (DAC). According to experts on the Intergovernmental Panel on Climate Change (IPCC), the massive deployment of BECCS technology represents a cost-effective strategy [2]. The International Energy Agency (IEA) considers that BECCS could play an important role in the decarbonization of the power sector, contributing with 5% of total CO$_2$ emission reduction in this sector by 2070 [3]. There are different BECCS technologies classified as a function of the biomass

conversion pathways as follows: thermochemical, mechanical/chemical, thermo, and bio-chemical [4]. Among these options, the thermochemical and biological are the most used at commercial scale [1]. It is important to mention that biomass is abundant, especially in most developing countries. For example, biomass such as bamboo, rice, maize, sugarcane, sorghum, and wheat represent 85% of the total amount of residue produced in Mexico [5,6].

On the other hand, the importance of $H_2$ as an energy vector has increased in recent years due to its widespread use and versatility e.g., in fuel cell vehicles, electricity, heating, and industrial feedstock; therefore, $H_2$ has been considered an essential fuel for a decarbonized world [7–11]. Steam methane reforming (SMR) is the most used process to produce $H_2$ due to its technological maturity and economical production at commercial size [8,10]. Despite the economic benefits of SMR over other processes, it has a large carbon footprint [8,12]. By stoichiometry, 5.5 kg of $CO_2$ are produced per kg of $H_2$ [13]. One option to reduce its carbon footprint is the implementation of CCUS technology. The $H_2$ obtained through an SMR unit with CCS or CCUS is regarded as blue $H_2$ ($bH_2$). It can help to meet climate change goals at acceptable costs [7,8]. However, the deployment of $bH_2$ is not necessarily $CO_2$-free. $CO_2$ capture efficiencies are expected to reach 85–95% at best, which means that 5–15% of all $CO_2$ is leaked [9,11].

One of the promising technologies to produce clean $H_2$ is the electrolysis process with surpluses of renewable energy (e.g., wind, solar), also known as "power-to-gas" (P2G) technology. The gas produced is called green $H_2$ ($gH_2$) due the fact that the electricity consumed to produce it comes from a carbon-free process. However, the electrolysis process requires a high amount of electricity. The power required to generate 1 kg of $H_2$ is approximately 60.6 kWh, assuming an efficiency of 65% of the electrolyzer system, which leads to it being considered a commercially available technology [8,13]. A cleaner version than $gH_2$ is net negative emission $H_2$, which is produced using BECCS technology.

The negative emission $H_2$ is commonly produced via pyrolysis and gasification processes, transforming the biomass chemically into syngas, which is a fuel gas mixture consisting primarily of $H_2$, CO, and $CO_2$. However, the presence of $N_2$ and $CO_2$ in the syngas contributes to the reduction of the lower heating value (LHV), which reduces the heat capacity of the syngas and in turn, limits its use in conventional power plant technologies and future technologies e.g., fuel cells. For example, syngas produced from biomass in a gasifier has a low content of $H_2$ and $CH_4$, which leads to an LHV of 5.6 MJ·(Nm)$^{-3}$ [14]. An internal combustion engine requires a minimum LHV of 4.6 MJ·m$^{-3}$ [15] and micro-turbines require 13.04 MJ·m$^{-3}$ [16]. In the case of $H_2$ utilization on fuel cell applications, $H_2$ purity varies depending on the type of fuel cell used. For example, in the case of proton exchange membrane (PEM) fuel cell applications, $H_2$ concentration in the fuel stream must be almost pure (≤99.97%) in order to avoid damage of the power device, so its use and market is limited to $gH_2$. Meanwhile, the solid oxide fuel cells (SOFC), in addition to operating with $H_2$, can work with different types of fuels such as synthesis gas, natural gas, and methanol [17] due to their higher operating temperatures than PEM fuel cells (100–300 °C). Generally, the SOFC operates in the region of 600 to 1000 °C. This means that high reaction rates can be achieved without expensive catalysts [18], as is currently necessary for lower temperature fuel cells (PEM), and SOFCs are not vulnerable to carbon catalyst poisoning. Among the SOFC types, planar SOFC (PSOFC) and tubular (TSOFC) geometry are the most promising because high-temperature gas-tight seals are eliminated, but one of the major disadvantages of the planar design is the need for gas-tight sealing around the edge of the cell components [18].

For $H_2$ purification by $CO_2$ capture, adsorption technologies have been widely investigated, considered as the alternative for the amine-based absorption carbon capture, and are a nearly mature technology [19]. Solid adsorption capture uses solid sorbents that are easy to control, require low regeneration heat and low capital investment. Adsorption processes can generally be classified into two types i.e., pressure swing adsorption (PSA

and temperature swing adsorption (TSA) [20]. For PSA, adsorption is performed at pressures higher than atmospheric whereas TSA is heated by a feed of hot steam. When the adsorption step is performed at atmospheric pressure or lower, PSA is termed VPSA. PSA is commonly used for $H_2$ production [21]. Liu et al. [22] investigated a two-stage VSA/PSA process for $CO_2$ capture and $H_2$ production from an SMR gas mixture. Results indicated the higher performance of PSA over their previous work i.e., single-stage PSA process. Wassie et al. [23] conducted a detailed thermodynamic and economic analysis of novel membrane-assisted gas switching reforming (MA-GSR). Results showed that the MA-GSR process achieved a similar $H_2$ production cost as conventional SMR without $CO_2$ capture.

Several research studies have been carried out to evaluate different biomass-fuelled power plant systems [24]. Zhen et al. [25] investigated the gasification performance of key components, including polyethylene and bamboo of municipal solid waste in a bench-scale fixed bed. It was observed that an optimal temperature for bamboo is 700 °C for the best syngas quality and the highest LHV is 6.22 MJ·Nm$^{-3}$. Chiang et al. [26] tested gasification by using bamboo chopsticks as the feedstock. The syngas heating value was significantly enhanced by CaO additions, which was related to the higher reaction rate of water–gas shift reaction. Usach et al. [27] studied a farm-based biogas fueled trigeneration system with different cooling pathways. Results indicated that none of the pathways increased the economic viability of the plant due to low electricity prices.

Based on the different challenges to decarbonize and meet environmental targets, the novelty of this paper is the integration of gasification with PSA using activated carbon (AC) to purify the produced $H_2$. Then for generating heat and electricity, a tubular solid oxide fuel cell (TSOFC) is integrated, due to its advantages over other type of fuel cells as described above. SOFC can convert CO and $H_2$, and the high operating temperature allows internal reforming of gaseous fuel and the production of high quality heat for the cogeneration system, which definitely increases the efficiency of the system. The framework of this paper is illustrated as follows: methodology for wood gasification, $H_2$ purification, and fuel cell simulations is presented in Section 2. Then, results of the simulation process and $CO_2$ emission analysis are presented in an integrated system described in Section 3. Finally, a conclusion on the potential of the proposed system is reached.

## 2. Methodology

### 2.1. Process Description

Figure 1 shows the process flow diagram of the integrated bioenergy-fuel cell cogeneration plant simulated in Aspen Plus. This consists of three main processes: (a) syngas production from wood gasification, (b) $H_2$ purification using a PSA unit; and (c) power and heat generation through a tubular solid oxide fuel cell (TSOFC) stack. More details on the modeling of each process are given below.

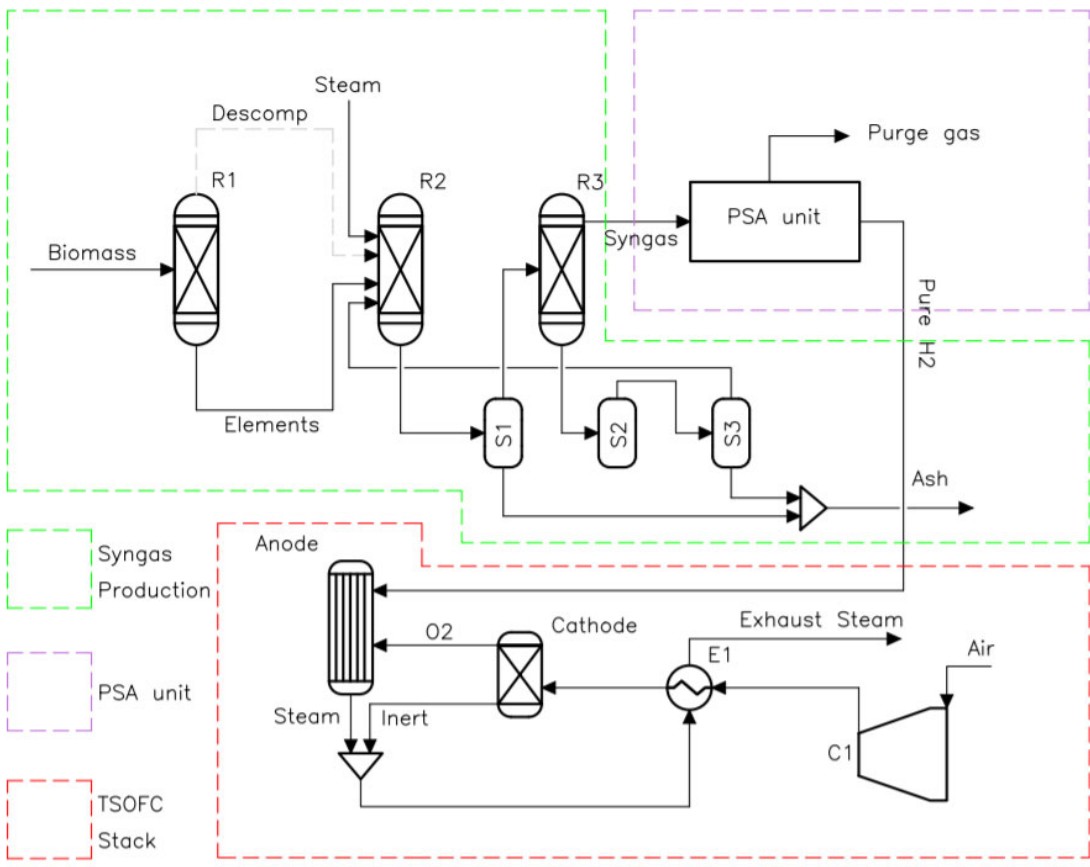

**Figure 1.** Process flow diagram of the integrated bioenergy-fuel cell cogeneration plant simulated in Aspen Plus.

### 2.1.1. Syngas Production from Wood Gasification

The syngas production process is modeled using Aspen Plus® software. The modelling process consists of four physicochemical steps: decomposition, pyrolysis, gasification, and separation. The amount of wood that is fed into the gasifier is 132 kg/h. Wood biomass feedstock is a non-conventional material in Aspen library (*Aspen Plus V11*; Aspen, CO, USA, 2021), which cannot take part in the thermodynamic chemical reactions. Hence, the Ryield reactor (R1) represents decomposition, where biomass is first broken-down into its elemental components e.g., ash, $H_2$, $N_2$, S, $O_2$, C, the composition depends on the type of wood or biomass used. This type of reactor is recommended when both the reaction kinetics and stoichiometry are unknown. Table 1 shows the thermodynamic input data used in the Ryield reactor (R1) model, which is based on the work developed by Islam [28]. The yield percentage is specified into the model using an inbuilt calculation block via Fortran statement. Therefore, the products of this reactor are the percentage elemental composition of the biomass which is subsequently fed into the first Gibbs reactor (R2) where pyrolysis, combustion, and gasification reactions occur, reported in Table 2 according to the reference [26]. When using air as gasifying agent, the ratio of air to biomass is indicated with the parameter equivalence ratio (ER). In this work, steam was used as the gasifying agent to increase the LHV of the syngas. A separator (S1) is placed after the R2 reactor to separate the products into syngas and ash. A second Gibbs reactor (R3) is positioned after the separator to adjust the composition of the synthesis gas. The product is passed to the second separator (S2) where the remaining solids and entrainment gas from the gas synthesis are separated. Finally, a third separator (S3) extracts the remaining entrained gas

from the solids, which is returned to R2. The ash from each separator is collected for disposal in landfill.

**Table 1.** Thermodynamic input data used in the Ryield reactor model. Capacity of the gasifier is 132 kg/h of biomass.

| Ultimate Analysis (wt.%, Dry) | Wood Chip (Islam, 2020) [28] |
|---|---|
| Ash | 0.450 |
| Carbon | 52.463 |
| Hydrogen | 7.466 |
| Nitrogen | 0.100 |
| Sulfur | 0.299 |
| Oxygen | 39.223 |
| Proximate analysis (wt.%, dry) | |
| Fixed carbon | 17.15 |
| Volatile matter | 82.4 |
| Ash | 0.45 |
| Moisture | 13.48 |
| Low heating value (LHV) (MJ/kg) | 19.54 |
| ER | 0.25 |
| Temperature (ºC) | 827 |

**Table 2.** Reactions in each step of biomass gasification.

| | | |
|---|---|---|
| **Pyrolysis** | endothermic stage | Biomass $\leftrightarrow$ $H_2$ + CO + $CO_2$ + $CH_4$ + $H_2O(g)$ + Tar + Char |
| **Oxidation** | exothermic stage | $C + O_2 \rightarrow CO_2$ Char combustion |
| | | $C + \frac{1}{2}O_2 \rightarrow CO$ Partial oxidation |
| | | $H_2 + \frac{1}{2}O_2 \rightarrow H_2O$ Hydrogen combustion |
| **Reduction** | endothermic stage | $C + CO_2 \leftrightarrow 2CO$ Boudouard reaction |
| | | $C + H_2O \leftrightarrow CO + H_2$ Reforming of the Char |
| | | $CO + H_2O \leftrightarrow CO_2 + H_2$ Water shift reaction |
| | | $C + 2H_2 \leftrightarrow CH_4$ Methanation |

For the modeling process, the Peng-Robinson with the Boston-Mathias modifications (PR-BM) state equation was selected because of its ability to calculate the thermodynamic properties of the participating fluids at the operating temperature and pressure. Other assumptions considered in the simulation are as follows:

- The system is isothermal and operates under steady-state conditions without transients.
- Pressure drops are overlooked. The formation of tars is neglected.
- The composition of the char is 100% carbon.
- The process is carried out under atmospheric pressure.
- Heat losses from the gasifier are ignored.

### 2.1.2. $H_2$ Purification Using a PSA Unit

The syngas contains a high percentage of $H_2$, which contains other components e.g., $CO_2$, CO, $N_2$. Thus, in order to use it in a fuel cell, as mentioned previously, it must be purified, in this work in a PSA. Figure 2 shows the schematic diagram for the cycle sequence used in PSA simulations. For the cycle configuration, two cycles are a sequence of adsorption-regeneration steps, which are important processes for cyclic stability. The room temperature 25 °C is considered for the PSA process. Activated carbon is selected as the adsorbent. The average mass flow rate of $H_2$ is 10.57 kg/h when S/B are 0.3 and 0.4. Equations (1) and (2) describe the Dual Langmuir isotherm and Arrhenius equation used

for PSA for $H_2$ purification. Recovery rate and productivity are evaluated based on Equations (3) and (4). More detailed information can be found in reference [29].

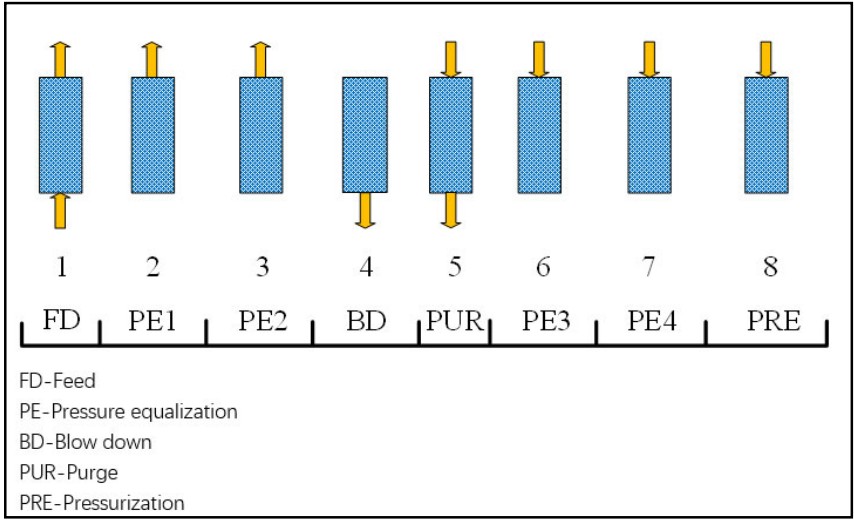

**Figure 2.** Cycle sequence used in the pressure swing adsorption (PSA) simulations.

$$q_i = \frac{q_{max1,i} b_{1,i} p_i}{1 + b_{1,i} p_i} + \frac{q_{max1,i} b_{1,i} p_i}{1 + b_{1,i} p_i} \tag{1}$$

where $q_i$ is the adsorption capacity, mol·kg$^{-1}$; $q_{max1,i}$ and $q_{max2,i}$ are the maximum capacity, mol·kg$^{-1}$; $b_{1,i}$ and $b_{2,i}$ are the adsorption constant, Pa$^{-1}$. The values for the parameters of AC adsorption are as follows: $q_{max1}$ is 1.79 mol·kg$^{-1}$; $b_{01}$ is 1.33 × 10$^{-8}$ Pa$^{-1}$.

$$b_i = b_{0i} \exp\left(-\frac{\Delta H_{ads,i}}{RT}\right) b_i = b_{0i} \exp\left(-\frac{\Delta H_{ads,i}}{RT}\right) \tag{2}$$

where $b_{0i}$ is the pre-exponential factor and $\Delta H_{ads,i}$ is the adsorption enthalpy which is 5926 J·mol$^{-1}$.

$$Recovery\ rate(\%) = \frac{mol_{H2,feed} - mol_{H2,pro}}{mol_{H2,feed}} \tag{3}$$

$$Productivity\ (mol_{H2}/kgs) = \frac{mol_{H2,feed} - mol_{H2,pro}}{kg_{AC,feed} t_{cycle}} \tag{4}$$

2.1.3. Power and Heat Generation through a TSOFC

The model simulation of the TSOFC stack process was based on the work developed by Tanim and coworkers [30], who assumed the following considerations: zero-dimensional; isothermal and steady state operation; all working fluids are treated as ideal gases; pressure drops are neglected. The zero-dimensional model generally assumes that chemical and thermodynamic equilibrium is present at the output streams [31].

Each TSOFC unit consists of two main components: one cathode and one anode. In general, a compressed and preheated air stream is fed to the cathode, which separates around 90% of the $O_2$ contained in the air by an electrochemical process. Then, the purified $O_2$ stream is fed to the anode, where it reacts with the $H_2$, generating power and steam; this last is fed to a heat exchanger for air preheating. Since the steam is still hot (128 °C), it can be used for other heating purposes. The ANODE is simulated as an equilibrium Gibbs reactor with an operating temperature and pressure equal to 3 bar and 850 °C, respectively. The CATHODE is also simulated as an equilibrium Gibbs reactor but with the difference that includes a Fortran routine to calculate the actual operating voltage and the

gross electric power of the TSOFC. The actual operating voltage of the cell ($V_c$) can be determined as follows [30]:

$$V_c = V_N - (n_{ohm} + n_{act} + n_{con}) \tag{5}$$

$$V_N = -\frac{\Delta \bar{g}_f}{2*F} + \frac{R_g * T_{avg}}{2*F} ln \frac{P_{H_2} * P_{H_2}^{0.5}}{P_{H_2O}} \tag{6}$$

where $V_N$ is the Nerst's voltage, in Volts; $\Delta \bar{g}_f$ is the molar Gibbs free energy of formation in J/mol; F is the Faraday constant (96,485 C/mol); $T_{avg}$ is the average temperature of the TSOFC inlet and outlet streams (K); $P_i$ is the partial pressure (bar) of the gaseous species. The variables $n_{ohm}$, $n_{act}$, and $n_{con}$ are the ohmic, activation, and concentration losses in the cell, respectively (V). In this work, the $n_{ohm}$ and $n_{act}$ were calculated using the expressions proposed by Song and coworkers [32], and the geometry and materials properties published by Siemens-Westinghouse [30,32,33], resulting in values equal to 0.089 V and 0.18 V, respectively. $n_{con}$ is neglected because the diffusion at high operating temperature in the TSOFC is a very efficient process [34]. In turn, the gross electric power ($P_{el}$), defined in Watts, is calculated as follows [18]:

$$P_{el} = 4F * n_{O_2} * V_c \tag{7}$$

where $n_{O_2}$ corresponds to the moles of oxygen consumption in the fuel cell. Finally, the gross electrical efficiency of the TSOFC can be expressed as follows [30]:

$$\eta_{el,gross} = \frac{P_{el,AC}}{\dot{\eta}_{fuel} * LHV_{fuel}} \tag{8}$$

where $P_{el,AC}$ is the electrical AC power (W) after converting from DC power. In this work an inverter efficiency factor equal to 92% is used [35]; $\dot{\eta}_{fuel}$ is the mass flow rate of $H_2$ in kg/s and $LHV_{fuel}$ is the lower heating value of $H_2$ (120 kJ/kg).

## 3. Results and Discussion

The model of the gasifier process is validated against the experimental and simulation results reported by Islam [28] to provide confidence in this work. Table 3 presents the comparison results of the syngas thermodynamics properties obtained with the reference data in Table 1. It is observed that the composition of the CO and $CO_2$, as well as LHV, are in the range between two data sets, which is an improvement from the simulation results of Islam [28]. Discrepancies between the experimental and this simulation results are mostly due to the assumption of chemical equilibrium, which yields the maximum attainable component conversions. Table 3 also shows in brackets the sum squared deviation (RSS) used to estimate the accuracy of the simulation results [36]. The syngas in this work contains 26.64% of $H_2$ and 1.18% of $CH_4$.

**Table 3.** Comparison results of the syngas thermodynamics properties obtained in this work with the reference data.

| Wood Chip | Reference (Islam, 2020) [28] Experimental | Reference (Islam, 2020) [28] Aspen Plus Simulation (RSS) [c] | This work [a] Aspen Plus Simulation (RSS) [c] |
|---|---|---|---|
| $H_2$ (% mol) | 8 | 13 (0.39) | 26.64 (5.43) |
| $N_2$ (% mol) | N.A. [b] | N.A. [b] | 36.10 |
| $H_2O$ (% mol) | N.A. [b] | N.A. [b] | 5.90 |
| CO (% mol) | 19.5 | 22.5 (0.02) | 21.52 (0.01) |
| $CO_2$ (% mol) | 6.5 | 11 (0.48) | 8.57 (0.10) |
| $CH_4$ (% mol) | 4 | 11 (3.06) | 1.18 (0.49) |

| Low heating value (LHV) (MJ/Nm³) | 4.93 | 8.48 (0.52) | 6.24 (0.07) |
|---|---|---|---|
| Carbon conversion efficiency | N.A. [b] | N.A. [b] | 0.935 |

[a] Thermodynamic method: PR-BM, Gasifier: RGibbs block with Restricted chemical equilibrium (temperature approach for the entire system −150 °C). [b] N.A. = not available, [c] RSS = sum squared deviation.

Due to the low LHV value of the syngas when using air as the gasifying agent presented in Tables 4 and 5 shows the results of the syngas thermodynamics properties where steam to biomass (S/B) ratio is varied. As seen in Table 4, the maximum $H_2$ concentration is reached for a S/B ratio equal to 0.4, corresponding to a $H_2$ composition equal to 49.09% mol, with an LHV value of 9.39 MJ/Nm³. Additionally, the highest LHV corresponds to a S/B ratio equal to 0.3 with 10.12 MJ/Nm³. The carbon conversion efficiency increases as a function of the S/B ratio due to more available oxidant. Mass and energy balance is presented in Appendix A.

**Table 4.** Syngas thermodynamics properties as a function of the S/B ratio.

| S/B Ratio | 0.3 | 0.4 | 0.6 |
|---|---|---|---|
| $H_2$ mol% | 48.99 | 49.09 | 47.85 |
| $N_2$ mol% | 0.03 | 0.03 | 0.02 |
| $H_2O$ mol% | 12.13 | 14.81 | 21.73 |
| CO mol% | 24.05 | 21.43 | 15.94 |
| $CO_2$ mol% | 10.71 | 11.63 | 13.01 |
| $CH_4$ mol% | 3.99 | 2.93 | 1.38 |
| LHV (MJ/Nm³) | 10.12 | 9.39 | 7.96 |
| Carbon conversion efficiency | 0.961 | 1.0 | 1.0 |

As in Tables 4 and 5 shows the gas composition, after $H_2$ is removed, at different S/B ratios. The purity of the $H_2$ obtained is 99.99% and the rest of the gas consists basically of CO, $CO_2$, and $H_2O$, where CO is the most abundant component around 51.8–58.9%. Although the increment of the ratio S/B reduces the amount of $N_2$, and increases the content of $H_2O$, it is important to note that the work duty required for $H_2$ separation in the PSA process decreases from 7.02 kW at S/B = 0.3 to 6.98 kW at S/B = 0.4 when flow rates of hydrogen are 10.57 kg/h and 10.54 kg/h, respectively.

**Table 5.** Simulation results of the PSA at different S/B ratios, and $H_2$ purity = 99.99 mol%.

| PSA results | Units | S/B | |
|---|---|---|---|
| | | 0.4 | 0.3 |
| **Composition in the Purge Gas** | | | |
| $H_2$ | mass fraction | 0.0709 | 0.0744 |
| $O_2$ | mass fraction | $9.747 \times 10^{-19}$ | $5.499 \times 10^{-19}$ |
| $N_2$ | mass fraction | 0.000537 | 0.000601 |
| $H_2O$ | mass fraction | 0.1874 | 0.1464 |
| CO | mass fraction | 0.5181 | 0.5888 |
| $CO_2$ | mass fraction | 0.2208 | 0.1869 |
| $CH_4$ | mass fraction | 0.00055 | 0.00089 |
| $H_2S$ | mass fraction | 0.0017 | 0.0019 |
| $H_3N$ | mass fraction | $2.108 \times 10^{-6}$ | $2.354 \times 10^{-6}$ |
| S | mass fraction | $8.459 \times 10^{-12}$ | $9.176 \times 10^{-12}$ |
| Flow rate of $H_2$ | kg/h | 10.57 | 10.54 |

| | | | |
|---|---|---|---|
| Work capacity | kJ/mol$_{H2}$ | 4.76 | 4.80 |
| H$_2$ recovery | % | 75.1 | 74.3 |
| Work capacity | kW | 6.98 | 7.02 |

Table 6 shows the results of the fuel cell TSOFC system. The overall electrochemical reaction in the anode took place at 3 bar and 850 °C, with an H$_2$ feed of 0.0029 kg/s that reacted stoichiometrically with O$_2$ to produce 1.11 V at an average temperature of 425 °C; however, activation and ohmic losses (anode, cathode, electrolyte, and interconnector) were discounted, leaving a cell voltage of 0.831 V. The tubular cell produced an AC power output of 217.1 kW with a net electrical efficiency of 58.2%, and as byproducts steam (0.0272 kg/s) and usable heat (100 kW). After comparing this work with the Mitsubishi Hitachi Power Systems, where the cell voltage changed by 86.76% and 8.93 % in net electrical efficiency, the Mitsubishi showed lower cell voltage due to high cell packing that allowed a higher current density. When using the Siemens–Westinghouse system, the change was 26.59% for voltage and 2.06% in efficiency; the voltage change is mainly due to the level of hydrogen purity since H$_2$ is produced by hydrocarbon reforming. Nonetheless, Siemens–Westinghouse's cell is integrated with a gas turbine that generates additional power and improves efficiency.

**Table 6.** Simulation results of the fuel cell tubular solid oxide fuel cell (TSOFC) system.

| TSOFC Results | Unit | This work | Mitsubishi Hitachi Power Systems MEGAMIE [37] | Siemens–Westinghouse [32,33] |
|---|---|---|---|---|
| Cell operating temperature | °C | 850 | 850 | 850 |
| Cell voltage | V | 0.831 | 0.11 | 0.61 |
| AC Power output | kW | 217.1 | 210 | 220 |
| Net electrical efficiency | % | 58.2 | 53.0 | 57.0 |
| Power consumption of air compressor | kW | 12.12 | - | - |
| Discharge air pressure | bar | 2.3 | - | - |
| Flue gas mass flow | kg/s | 0.1165 | N.A. | N.A. |
| H$_2$ fuel consumption | kg/s | 0.0029 | N.A. | N.A. |
| Supplementary fuel | kg/h | 0.80 | | |
| Steam generated | kg/h | 60 | | |

N.A. = not available.

The 132 kg/h of biomass produces 10.57 kg/h or 0.0029 kg/s of H$_2$ in the gasifier, after the PSA, which is fed to the fuel cell. The fuel cell generates 217 kW of electricity and the exhaust gas so that fuel cell has enough heat to generates 40 kg/h of steam needed in the gasifier. In order to complete 60 kg/h, 0.8 kg/h of natural gas is needed to burn as supplementary firing, as shown in Table 6.

Table 7 summarizes the key information of the integrated system (gasification, PSA, and fuel cell) used to estimate the efficiency of the integrated system (gasification process, PSA, and TSOFC).

**Table 7.** Summary of key information of the integrated system.

| Concept | Unit | Amount |
|---|---|---|
| Biomass mass flow | kg/h | 132 |
| LHV of biomass | kJ/kg | 19540 |
| LHV natural gas | kJ/kg | 50047 |
| Supplementary gas | kg/h | 0.8 |
| H$_2$ produced | kg/h | 10.57 |

| | | |
|---|---|---|
| AC gross power output | kW | 217.1 |
| Total work duty required for $H_2$ separation | kW | 6.98 |
| Total word required to pump the water for steam in the fuel cell | kW | 0.084 |
| Power consumption of air compressor | kW | 12.12 |
| Net power | kW | 197.92 |
| Net efficiency | % | 27.20 |
| $CO_2$ generated in the gasifier | kg/h | 35.41 |

The efficiency of the integrated system is estimated based on the information presented in Table 7 by using the following equation:

$$\eta_{en} = \frac{W_p}{m_s \, LHV_{H2} + m_{NA} \, LHV_{NG}} * 100\% \tag{9}$$

where $\eta_{en}$ is the net efficiency of the integrated system in %; $W_p$ is the net power output generated in the fuel cell TSOFC in kW (This power considers); $m_s$ is the mass flow rate of the wood biomass in kg/s$^{-1}$; LHV$_{H2}$ is low heating value of biomass in kJ/kg$^{-1}$; $m_{NA}$ is the mass flow rate of the supplementary natural gas in kg/s$^{-1}$, LHV$_{NG}$ is low heating value of natural gas in kJ/kg$^{-1}$.

The efficiency of the integrated system is estimated as 27.2%, a value lower than that of a gas turbine with the same capacity whose efficiency is around 33.1% [38]. Another alternative method for generating power with biomass is a microturbine. However, according to the manufacturer Capstone, the minimum limit of LHV established to be used in the Capstone C200 micro-turbine is 13.04 MJ/Nm$^{-3}$, but the syngas generated in this study is around 10.12 MJ/Nm$^3$ at S/B = 3. Then, it would be necessary to mix with different percentages of natural gas or liquid petroleum gas (LPG) in order to improve the LHV of the syngas; this could be a disadvantage for rural and remote areas because of the unavailability of these fossil fuels. Nonetheless, the use of fossil fuels would increase $CO_2$ content, which is an advantage for fuel cells. In addition, if the steam generated in the fuel cell could be used to generate power in a small steam turbine or for thermal heating that could improve the efficiency of the system presented in this work even more.

Total $CO_2$ emitted by the integrated system presented in this work is estimated considering the $CO_2$ emitted and the $CO_2$ sequestered by wood. The $CO_2$ sequestered by wood is estimated by the simple equation 10:

$$CO_2 \text{ per tree} = [\text{Tree mass} \times 0.65 \times 0.50 \times 1.20 \times 3.67]/[\text{age of biomass}] \tag{10}$$

where $CO_2$ per tree is the sequestered $CO_2$ in kg/h; tree mass is the mass flow of fresh biomass in kg/h; 65% is the percentage of dry mass; 50% is the percentage of carbon. As 20% of tree biomass is below ground level in roots, the equation is multiplied by a factor of 120%. Finally, the equation is multiplied by 3.67, which is the ratio of $CO_2$ to C: 44/12 = 3.67.

Considering a 12-year old tree, total annual mass of tree is around 1,056,000 kg per year (132 kg/h, considering 8000 h of operation during the year). Using Equation (10), total $CO_2$ absorbed per year is 125,954 kg/y. $CO_2$ generated in the gasifier per year is 283,360 kg/y. Total net $CO_2$ emitted by the system is 157,406 kg/y or 19.67 kg/h, and the $CO_2$ intensity is 98.3 kgCO$_2$/MWh. Compared with a NGCC without and with carbon capture, their carbon intensities are around 331 kgCO$_2$/MWh and 40 kgCO$_2$/MWh, respectively. An economic analysis is needed to compare the proposed integrated system with an NGCC with $CO_2$ capture. However, this is beyond the scope of this work and will be presented as future work.

## 4. Conclusions

This study demonstrates that it is possible to combine biomass gasification and fuel cell technologies as an alternative option to generate clean electricity for remote areas. This concept was tested by using wood to generate a $H_2$-rich gas in a gasifier. The integrated system could generate a net power of 197.92 kW and 60 kg/h of steam (used in the gasifier) when using 132 kg/h of wood. The net efficiency of the integrated system, considering only the electric power generated in the TSOFC, is lower than a gas turbine with the same capacity: 27.2% against 33.1%.

The carbon intensity of the system presented in this work is 98.3 kgCO$_2$/MWh compared with those of NGCC without and with CO$_2$ capture, i.e., 331 kgCO$_2$/MWh and 40 kgCO$_2$/MWh, respectively. If CCUS is integrated in the system, power with negative carbon emission is possible.

There is great potential for cogeneration with low carbon emission by using wood biomass in rural areas of developing countries. This is an alternative technology for places where biomass is abundant and where it is difficult to get electricity from the grid due to geographical location limits.

**Author Contributions:** Conceptualization, A.G.-D.; methodology, A.G.-D., C.F.-P., P.D.-H., M.O.G.-D.; software, A.G.-D., C.F.-P., L.J., and J.C.S.L.d.G.; validation, C.F.-P., L.J., J.C.S.L.d.G.; formal analysis, A.G.-D., C.F.-P., L.J.; investigation, J.C.S.L.d.G.; data curation, A.G.-D., C.F.-P., and J.C.S.L.d.G.; writing—original draft preparation, A.G.-D., C.F.-P., L.J., P.D.-H.; M.O.G.-D.; writing—review and editing, A.G.-D., C.F.-P., L.J., P.D.-H.; M.O.G.-D.; visualization, M.O.G.-D.; supervision, A.G.-D. and C.F.-P.; project administration, C.F.-P.; funding acquisition, C.F.-P. All authors have read and agreed to the published version of the manuscript.

**Funding:** This research received no external funding.

**Institutional Review Board Statement:** Not applicable.

**Informed Consent Statement:** Not applicable.

**Data Availability Statement:** Not applicable.

**Acknowledgments:** The authors would like to thank Sufyan Aslam Mukadam for his enthusiastic support in the literature search related to biomass gasification and fuel cells. González-Diaz would like to thank the National Institute of Electricity for the support. Font-Palma was supported by the Royal Academy of Engineering under the Leverhulme Trust Research Fellowship scheme.

**Conflicts of Interest:** The authors declare no conflict of interest.

## Nomenclature

| | |
|---|---|
| AC | Activated carbon |
| BECC | Bioenergy with carbon capture and sequestration |
| CBC | Coastal blue carbon |
| CCS | carbon capture and storage |
| CCUS | carbon capture use and storage |
| DAC | Direct air capture |
| ER | equivalence ratio |
| GHG | greenhouse gas |
| HHV | higher heating value |
| IEA | International Energy Agency |
| LHV | Lower heating value |
| LPG | Liquid petroleum gas |
| MA-GSR | membrane-assisted gas switching reforming |
| NET | negative emissions technologies |
| NGCC | Combined cycle power plant |
| NG | Natural gas |
| PEM | proton exchange membrane |
| SMR | steam methane reforming |

| | |
|---|---|
| SOFT | solid oxide fuel cells |
| PSA | pressure swing adsorption |
| PSOFC | planar SOFC |
| S/B | steam to biomass |
| TCRS | Terrestrial Carbon Removal and Sequestration |
| TSA | temperature swing adsorption |
| TSOFC | tubular solid oxide fuel cell |
| VPSA | vacuum pressure swing adsorption |

## Appendix A

Mass balance of gasifier

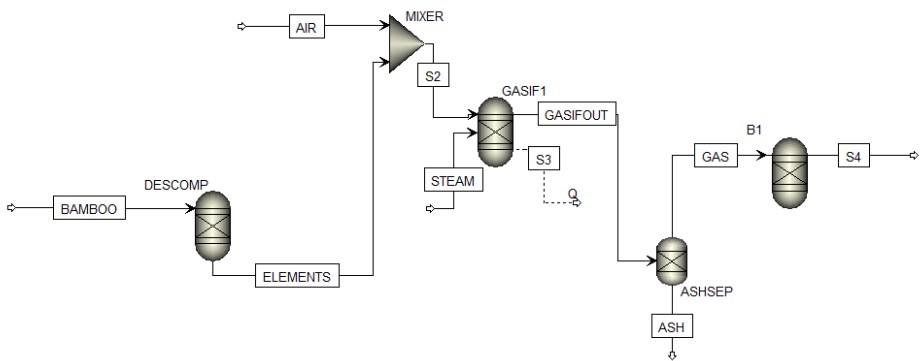

**Figure A1.** Aspen plus process diagram.

**Table A1.** Mass balance, temperature and pressure of the streams of the gasifier at B/S = 0.3.

| Stream Name | Units | ASH | BIOMASS | ELEMENTS | GAS | GASIFOUT | S2 | S4 | STEAM |
|---|---|---|---|---|---|---|---|---|---|
| Temperature | K | 1100.1 | 298.2 | 298.2 | 1100.1 | 1100.2 | 298.2 | 1123.2 | 423.2 |
| Pressure | atm | 1.04 | 1.00 | 1.04 | 1.04 | 1.04 | 1.04 | 1.04 | 1.04 |
| Mass Flows | kg/hr | 2.8 | 132.0 | 132.0 | 189.5 | 192.3 | 132.0 | 189.5 | 60.3 |
| Mole Fractions | | | | | | | | | |
| $H_2$ | | | | 0.6379 | 0.4899 | 0.4899 | 0.6379 | 0.5240 | 0.0000 |
| $O_2$ | | | | 0.2110 | 0.0000 | 0.0000 | 0.2110 | 0.0000 | 0.0000 |
| $N_2$ | | | | 0.0006 | 0.0003 | 0.0003 | 0.0006 | 0.0003 | 0.0000 |
| $H_2O$ | | | | 0.1489 | 0.1213 | 0.1213 | 0.1489 | 0.1154 | 1.0000 |
| CO | | | | 0.0000 | 0.2405 | 0.2405 | 0.0000 | 0.2984 | 0.0000 |
| $CO_2$ | | | | 0.0000 | 0.1071 | 0.1071 | 0.0000 | 0.0603 | 0.0000 |
| $CH_4$ | | | | 0.0000 | 0.0399 | 0.0399 | 0.0000 | 0.0008 | 0.0000 |
| $H_2S$ | | | | 0.0000 | 0.0009 | 0.0009 | 0.0000 | 0.0008 | 0.0000 |
| $H_3N$ | | | | 0.0000 | 0.0000 | 0.0000 | 0.0000 | 0.0000 | 0.0000 |
| S | | | | 0.0016 | 0.0000 | 0.0000 | 0.0016 | 0.0000 | 0.0000 |
| C | | | | 0.0000 | 0.0000 | 0.0000 | 0.0000 | 0.0000 | 0.0000 |
| $CL_2$ | | | | 0.0000 | 0.0000 | 0.0000 | 0.0000 | 0.0000 | 0.0000 |
| HCL | | | | 0.0000 | 0.0000 | 0.0000 | 0.0000 | 0.0000 | 0.0000 |

**Table A2.** Mass balance, temperature and pressure of the streams of the gasifier at B/S = 0.4.

| Stream Name | Units | ASH | BAMBOO | ELEMENTS | GAS | GASIFOUT | S2 | S4 | STEAM |
|---|---|---|---|---|---|---|---|---|---|
| Temperature | K | 1100.1 | 298.2 | 298.2 | 1100.1 | 1100.2 | 298.2 | 1123.2 | 423.2 |
| Pressure | atm | 1.04 | 1.00 | 1.04 | 1.04 | 1.04 | 1.04 | 1.04 | 1.04 |
| Mass Flows | kg/hr | 0.51 | 132.00 | 132.00 | 211.82 | 212.34 | 132.00 | 211.82 | 80.34 |
| Mole Fractions | | | | | | | | | |
| $H_2$ | | | | 0.6379 | 0.4909 | 0.4909 | 0.6379 | 0.5084 | 0.0000 |
| $O_2$ | | | | 0.2110 | 0.0000 | 0.0000 | 0.2110 | 0.0000 | 0.0000 |
| $N_2$ | | | | 0.0006 | 0.0003 | 0.0003 | 0.0006 | 0.0003 | 0.0000 |
| $H_2O$ | | | | 0.1489 | 0.1481 | 0.1481 | 0.1489 | 0.1503 | 1.0000 |
| CO | | | | 0.0000 | 0.2143 | 0.2143 | 0.0000 | 0.2673 | 0.0000 |
| $CO_2$ | | | | 0.0000 | 0.1163 | 0.1163 | 0.0000 | 0.0725 | 0.0000 |
| $CH_4$ | | | | 0.0000 | 0.0293 | 0.0293 | 0.0000 | 0.0005 | 0.0000 |
| $H_2S$ | | | | 0.0000 | 0.0008 | 0.0008 | 0.0000 | 0.0007 | 0.0000 |
| $H_3N$ | | | | 0.0000 | 0.0000 | 0.0000 | 0.0000 | 0.0000 | 0.0000 |
| S | | | | 0.0016 | 0.0000 | 0.0000 | 0.0016 | 0.0000 | 0.0000 |
| C | | | | 0.0000 | 0.0000 | 0.0000 | 0.0000 | 0.0000 | 0.0000 |
| $CL_2$ | | | | 0.0000 | 0.0000 | 0.0000 | 0.0000 | 0.0000 | 0.0000 |
| HCL | | | | 0.0000 | 0.0000 | 0.0000 | 0.0000 | 0.0000 | 0.0000 |

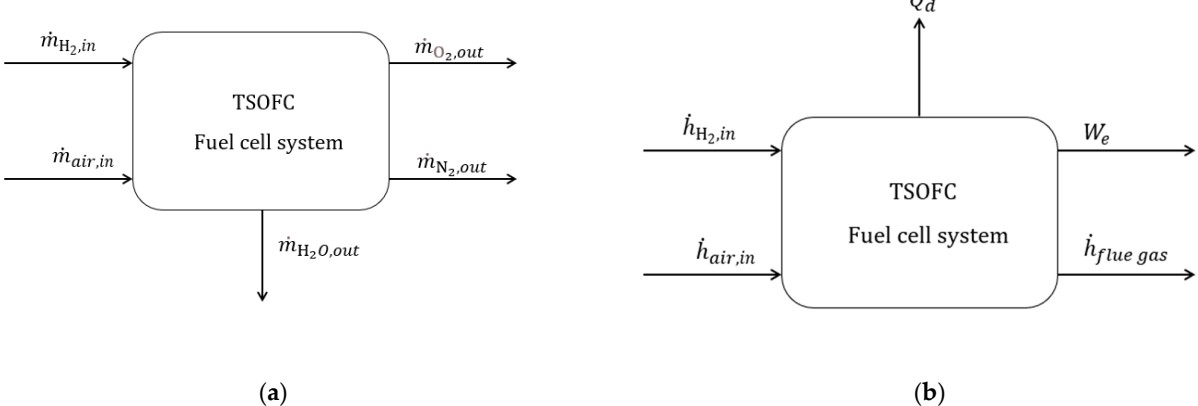

(**a**)          (**b**)

**Figure A2.** (**a**) Schematic mass balance for the TSOFC and (**b**) Schematic energy (enthalpy) balance for the TSOFC.

**Table A3.** Values for the mass and energy balance of TSOFC.

| Item | Unit | Inlet |
|---|---|---|
| Air mass flow | kg/s | 0.1133 |
| Air pressure | bar | 2.3 |
| Air temperature | °C | 320 |
| Air enthalpy flow | J/s | 25,073.6 |
| $H_2$ mass flow | kg/s | 0.0029 |
| $H_2$ pressure | bar | 2 |
| $H_2$ temperature | °C | 105 |
| $H_2$ enthalpy flow | J/s | 138.215 |
| Gas composition | | |
| $O_2$ | Mole fraction | 0.019 |
| $N_2$ | Mole fraction | 0.658 |
| $H_2O$ | Mole fraction | 0.322 |



| | | |
|---|---|---|
| Flue gas flow rate | kg/s | 0.116 |
| Flue gas pressure | Bar | 2 |
| Flue gas temperature | °C | 355.54 |
| Steam produced | kg/h | 45 |
| Steam pressure | bar | 3 |
| Temperature | °C | 300 |
| Power pump | kW | 0.0084 |

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
