# Peer review of "Techno-Environmental Analysis of the Use of Green Hydrogen for Cogeneration from the Gasification of Wood and Fuel Cell"

_sustainability, doi:10.3390/su13063232_

Round 1

Reviewer 1 Report

This work simulates the coupling of biomass gasification and solid oxide fuel cell for the low-carbon power generation in certain areas with abundant biomass resource. The model is well developed and validated, and properties of the integrated system including power output, energy efficiency and total CO2 emission are fully discussed and compared with other technologies. In below please find my comments and questions for its further improvement.

1) Biomass gasification coupled with fuel cell technology has already been studied extensively in literature. Instead of telling “the integration of a variety of technologies that could promote the use of green H2”, the authors need to better elaborate the novelty of their work against previous publications in the introduction section.

2) The introduction is also too lengthy. Suggest revising it in a more compact manner.

3) “H2 concentrations in the fuel stream must be almost pure (≤98%) in order to avoid damage of the power device”, I think the H2 purity for PEMFC should be more than 99.97% rather than only 98%.

4) In Table-3, the difference among the three columns is quite large in my opinion, especially for H2, CO2, CH4 and LHV. I wonder what is the standard for judging the level of agreement in this work. Is there any literature support?

5) For the calculation of total CO2 emission, it seems that there are some errors about the numbers. For example, the total CO2 absorbed calculated from Eq. 10 should be 1133360 rather than 94447 kg/y. Also, the numbers in kg/h and kg/h are not consistent with each other.

6) It is recommended to elaborate the abbreviations also in the abstract.

Author Response

For reviewer 1

Reviewer 2 Report

The manuscript entitled “Techno-environmental analysis of the use of green hydrogen for cogeneration from the gasification of wood and fuel cell” investigated the use of wood biomass in a gasifier integrated with a fuel cell system as a low carbon technology. It reveals that the net efficiency of the integrated system is higher than a conventional gas turbine with the same capacity. The authors made comprehensive analysis and clearly explained the integration of two systems. However, some small issues need to be addressed.

  1. Line 149-150, why do authors use the model including 5 steps (decomposition, pyrolysis, combustion, gasification and separation)? The syngas production from wood gasification only include pyrolysis and gasification steps. Please explain the details.
  2. What is ER in Table 1?
  3. Line 236-237. Authors claim that a good agreement is observed between the two data sets. It seems to be difficult to regard the data in Table 3 is in good correlations, especially comparing the H2 and CH4. More clarification are needed here.
  4. Mass and energy balance of gasification and TSOFC are needed.

Author Response

Reviewer 2

Reviewer 3 Report

Dear Authors,

Many thanks for your work. It is really good and well structured. Somebody could follow and get the total idea. It is really good that you present some values inside the abstract to give a clear idea to the readers. I have some general comments regarding your work, which I suggest you to add to them to create a clear understanding for the readers. My comments are listed as below:

1- Please recheck your work for the use of Abbreviations. Please check if they properly defined. I saw GHG is not defined. Even this must be defined properly.

2- I couldnt understand the novelty of your work! The main Idea is to integrate the systems! Otherwise, such things have been done properly by other researchers as well.

3- I suggest some tables become as figures to make it more visual.

4- Aspen Plus must be properly cited in the references!

5- Why you have used this specific wood and why you compared with Islams data! Due to similarity of the available biomass in your region! You know it could be changed region by region! 

6- Why dont you properly mentioned the end user of biomass gasification in real industry, which is highly followed by Sweden! If you know proper usage of this integration in power plant and Iron-steel industry is very of interest in Sweden as pioneer in this topic. In Sweden, such biomass derived syngas is tried to be applied in Iron and steel industry. For reduction and Also reheating Furnaces. I suggest you to rewrite introduction part to mention some real cases that is used in industry. For example I just give you a hint to a paper which uses the real case of the biomass derived syngas: "H. Liu, M. Saffaripour, P. Mellin, C.-E. Grip, W. Yang, W. Blasiak, A thermodynamic study of hot syngas impurities in steel reheating furnaces – Corrosion and interaction with oxide scales, Energy, Volume 77, 2014, Pages 352-361"

Author Response

Reviewer 3

Round 2

Reviewer 1 Report

Authors have well addressed most of my previous comments. However, I still have confusions on the calculation of total CO2 emission. If the total annual mass is around 1,056,000 kg/y, wouldn't the CO2 per tree be 1511453 kg/y rather than 125954 kg/y? Also, the CO2 generated in the gasifier of 283360 kg/y is not consistent with that in Table-7 (26.55 kg/h). Please correct me if I have missed any key information during the calculation.

Reviewer 3 Report

Dear Authors,

Many thanks for the revised version. I believe that the revised version is much more better than the initial draft. All my comments have been properly addressed. 

Author Response

Thank you so much for your comments and for your time and effort to revise the paper.